# In Vitro Induction of Eryptosis by Uremic Toxins and Inflammation Mediators in Healthy Red Blood Cells

**DOI:** 10.3390/jcm11185329

**Published:** 2022-09-10

**Authors:** Grazia Maria Virzì, Maria Mattiotti, Anna Clementi, Sabrina Milan Manani, Giovanni Giorgio Battaglia, Claudio Ronco, Monica Zanella

**Affiliations:** 1Department of Nephrology, Dialysis and Transplant, St Bortolo Hospital, 36100 Vicenza, Italy; 2IRRIV—International Renal Research Institute, 36100 Vicenza, Italy; 3Nephrology, Dialysis and Renal Transplant Unit, IRCCS—Azienda Ospedaliero-Universitaria di Bologna, Department of Experimental Diagnostic and Specialty Medicine (DIMES), Alma Mater Studiorum University of Bologna, 40126 Bologna, Italy; 4Department of Nephrology and Dialysis, Santa Marta and Santa Venera Hospital, 95024 Acireale, Italy

**Keywords:** eryptosis, cytokines, uremia, uremic toxin, red blood cells

## Abstract

Eryptosis is the stress-induced RBC (red blood cell) death mechanism. It is known that eryptosis is largely influenced by plasma and blood composition, and that it is accelerated in patients affected by chronic kidney disease (CKD). The aim of this study is to evaluate the eryptosis rate in healthy RBCs treated with different concentration of IL-6, IL-1β, urea and p-cresol, comparable to plasmatic level of CKD patients, at different time points. We exposed healthy RBCs to increasing concentrations of IL-6, IL-1β, urea and p-cresol. Morphological markers of eryptosis (cell membrane scrambling, cell shrinkage and PS exposure at RBC surface) were evaluated by flow cytometric analyses. The cytotoxic effect of cytokines and uremic toxins were analyzed in vitro on healthy RBCs at 4, 8 and 24 h. Morphology of treated RBCs was dramatically deranged, and the average cell volume was significantly higher in RBCs exposed to higher concentration of all molecules (all, *p* < 0.001). Furthermore, healthy RBCs incubated with each molecules demonstrated a significant increase in eryptosis. Cytofluorimetric analysis of eryptosis highlighted significantly higher cell death rate in RBCs incubated with a higher concentration of both cytokines compared with RBCs incubated with a lower concentration (all, *p* < 0.05). In conclusion, our data show that cytokines and uremic toxins have a harmful effect on RBCs viability and trigger eryptosis. Further studies are necessary to validate these results in vivo and to associate abnormal eryptosis with cytokine levels in CKD patients. The eryptosis pathway could, moreover, become a new promising target for anemia management in CKD patients.

## 1. Introduction

Erythrocytes (red blood cells, RBCs) are the most copious cells in the blood [1]. RBCs are enormously sensitive cells, with a highly specialized and organized membrane structure, which interacts and reacts to xenobiotic and endogenous elements [2,3,4]. Despite their powerful defense systems, the average life span is limited by senescence to 100–120 days. Eryptosis is stress-induced RBC death mechanism, distinct from accidental hemolysis or natural cellular senescence [2,3,4]. Eryptosis is a physiological mechanism which, through the removal of defective erythrocytes, prevents hemolysis and the subsequent release of hemoglobin into the circulating blood. Several stressors have been identified as triggers of eryptosis [4]. If accelerated eryptosis is not compensated by erythrocytes synthesis, anemia may occur. Eryptosis is characterized by cell shrinkage, cell membrane blebbing and scrambling with the exposure of the aminophospholipid phosphatidylserine (PS) on the outer surface of RBCs. PS-exposing RBCs may be quickly engulfed by macrophages, degraded and cleared from the circulation [5,6,7]. Moreover PS-exposing RBCs may bind the surface of endothelial cells, activating blood clotting and thus fostering thrombosis. The main molecular stimulus of eryptosis is increased intracytoplasmatic Ca^(2+)^ concentration, but oxidative stress, inflammation and uremic toxins also enhance eryptosis. Increased eryptosis has been observed in the pathogenesis of several clinical disorders [2,5,8,9,10,11,12]. In patients affected by chronic kidney disease (CKD), the life span of RBCs is further shortened [13]: the main contributor is the relative deficit of erythropoietin (EPO) [14,15], iron deficiency plays also a significant role, but it was recently proven that the erythrocytes of patients with end-stage renal disease (ESRD) show an increased percentage of PS externalization and subsequent eryptosis, if compared to healthy individuals [16].

The aim of this study was to evaluate the eryptosis level of healthy RBCs treated with different concentrations of urea, p-cresol, IL-6 and IL-1β, comparable to the plasmatic level of CKD and uremic patients at different time points. To the best of our knowledge, despite what was already proven for other cytokines and uremic toxins, there are no available studies about their effects on the RBC life span, in particular for IL-6, IL-1β, urea and p-cresol.

## 2. Materials and Methods

We exposed healthy whole blood to increasing concentrations of urea, p-cresol, IL-6 and IL-1β. Those concentrations were comparable to concentrations found in the plasma and serum of CKD and uremic patients [17,18,19,20,21,22,23,24,25]. Each experiment was repeated three times.

### 2.1. Healthy Volunteers Features and Blood Collection

Blood samples were obtained from three healthy individuals of ages ranging from 30 to 40 years. Healthy subjects were recruited in the Blood Center of St. Bortolo Hospital, and blood from these subjects was kindly provided by the blood bank of St. Bortolo Hospital. Each volunteer, with any condition that enhances eryptosis (iron deficiency, metabolic syndrome, diabetes, fever and dehydration, sepsis, hemolytic uremic syndrome, mycoplasma infection, myelodysplastic syndrome, phosphate depletion, malaria, sickle-cell anemia, β-thalassemia, glucose-6-phosphate-dehydrogenase-(G6PD)-deficiency, hereditary spherocytosis, paroxysmal nocturnal hemoglobinuria, and Wilson’s disease) was excluded. Blood was collected in one single 4.5 mL EDTA tube. This collection was performed by a medical doctor, and the handling of samples was performed under a highly strict aseptical condition, in order to prevent any contamination of the samples. The procedures were in accordance with the Helsinki Declaration. The protocol and consent form were approved by the Ethics Committee of St. Bortolo Hospital (n. 58/17). All patients were informed about the experimental protocol and the objectives of the study before providing informed consent and biological samples.

### 2.2. Concentration of Molecules

We dissolved the powder of urea, p-cresol and cytokines (Sigma, St. Louis, MO, USA) in a complete cell medium (RPMI 1640 with stable L-glutamine (International PBI Italy, Milan, Italy) and 10% fetal bovine serum (Sigma)). We obtained a final concentration of 2000 ng/µL for IL-6, 1000 ng/µL for IL-1β, 300 mg/L for urea, and 40 mg/L for p-cresol. Serial dilutions were made with the same medium (IL-6: 2000, 1000, 500, 250, 125, 62.5, 31.25, 0 ng/µL; IL-1β: 1000, 500, 250, 125, 62.5, 31.25, 15.63, 0 ng/µL, urea: 300, 250, 200, 150, 50, 0 mg/L and p-cresol: 40, 20, 10, 5, 2.5, 0 mg/L). 

### 2.3. In Vitro Exposure to Cytokines and Induction of Eryptosis

A total of 1.5 µL of whole EDTA blood was plated per well in 48-well plates in 300 µL of complete RPMI and incubated with increasing concentrations of selected molecules for 4, 8, and 24 h in RPMI 1640 under standard conditions (37 °C in 5% CO_2_). We used untreated RBCs as a negative control. We performed two separated experiments, and each concentration was tested twice.

### 2.4. Eryptosis Evaluation

The cell volume was determined with flow cytometry and forward scatter (FS). PS avidly binds annexin V, which was employed to identify eryptotic cells [7,26]. Then PS exposure on RBC surface was estimated from FITC-annexin V binding (Beckman Coulter, Brea, CA, USA) using Navios Flow Cytometer (Beckman Coulter, Brea, CA, USA). RBCs were gated and enumerated, identifying those cells that exposed PS at the RBC surface. A minimum of 100,000 events were collected on each sample.

## 3. Statistical Analysis

Statistical analysis was performed using the SPSS Software package. A *p*-value of <0.05 was considered statistically significant. Results are presented as percentages, or medians and interquartile ranges (nonparametric variables). The Mann–Whitney U test or T test was used for comparison of two groups when appropriate. The Kruskal–Wallis test or ANOVA test for multiple comparisons were applied to compare the groups when appropriate. Correlation coefficients were calculated with the Spearman’s rank or Pearson’s test, as appropriate.

## 4. Results

The cytotoxic effect of cytokines was evaluated in vitro on RBCs at 4, 8 and 24 h In order to investigate cell membrane scrambling, cell shrinkage and PS exposure on RBC surface as markers of eryptosis, eryptotic RBCs were identified by FS (cell volume dimension) and AnnexinV-binding with flow cytometric analyses. The morphology of RBCs incubated with higher concentration of IL-6, IL-1β, urea and p-cresol was dramatically deranged at each time point and for each concentration, if compared to untreated cells. The average FS, reflecting cell volume, was significantly higher in treated RBCs vs. untreated RBCs (*p* < 0.001) at any time point and for each concentration.

Eryptosis rate resulted in being significantly different between distinct concentration of molecules (for all molecules, *p* < 0.05). In particular, RBCs incubated with IL-6, IL-1β, urea and p-cresol demonstrated a significant increase in eryptosis at any time, compared to untreated cells (for all molecules, *p* < 0.05). Cytofluorimetric analysis of eryptosis highlighted a significantly higher cell death rate for RBCs incubated with the highest concentration of all substances if compared with cells exposed to a lower concentration and with untreated cells (*p* < 0.05). Eryptosis resulted in lower RBCs incubated with the lowest concentrations of all substances compared with other concentrations (*p* < 0.05). Very strong correlations between IL-6, urea and p-cresol concentrations and eryptosis level were observed (Table 1). Significant positive correlations between IL-1β concentrations and eryptosis level were observed. Figure 1, Figure 2, Figure 3 and Figure 4 show eryptosis induced by IL-6, IL-1β, urea and p-cresol in vitro at different time points (Figure 1, Figure 2, Figure 3 and Figure 4). Furthermore, we compared the effect of the same concentration of IL-6, IL-1β, urea and p-cresol at 4 h and at 24 h (Table 2), and eryptosis resulted in being increased after longer exposition (24 h) for all molecules (*p* ≤ 0.05).

Between tested molecules, we observed that IL-1β induced a lower rate of eryptosis for all concentrations and at all times if compared to other substances.

## 5. Discussion

In this study, we analyzed the in vitro induction of eryptosis in healthy RBCs treated with different concentration of uremic toxins and cytokines at different time points.

Anemia is a frequent complication during the later stages of CKD [13] and it is considered an independent risk factor of cardiovascular events and mortality, especially for patients with ESRD [14]. The pathogenesis of anemia is very complex: the main role is played by a relative deficit of EPO [15,27]; iron deficiency is the second main contributor [28]. Blood loss, infection, underlying hematologic disease, hyperparathyroidism, increased hemolysis and nutritional deficits have been recognized as additional factors in the pathogenesis [29]. In addition, the accelerated clearance of circulating erythrocytes [13,30], caused by enhanced eryptosis [6,7], also plays a crucial role.

Eryptosis is the premature, stress-induced death of red blood cells, comparable to the apoptosis of nucleated cells, which is distinct from accidental hemolysis or cellular senescence. Human erythrocytes are anucleate cells which display a life span of approximately 120 days, before being removed from circulation. The relatively long life span of these cells could be explained by two mechanisms: (i) during the last two stages of differentiation (i.e., reticulocytes and erythrocytes), cells extrude the nucleus and the mitochondrion, two organelles with a pivotal role in the apoptosis pathway, and (ii) erythrocytes express a high level of antiapoptotic proteins [31]. In contrast to classical hemolysis, which leads to destruction of plasma membrane and the subsequent release of hemoglobin into the extracellular space, the plasma membrane of eryptotic cells remains intact, and hemoglobin is retained in the cytoplasm [31].

The main trigger of eryptosis is the increased intracellular Ca^(2+)^ concentration [6,7] that enters through the Ca^(2+)^-permeable cation channel. The increase in cytosolic Ca^(2+)^ is followed by cell shrinkage, cell membrane scrambling with PS translocation to the cell surface [7]. Other stimulators of eryptosis include ceramide, energy depletion, caspase activation [32,33] and the deranged activity of kinases [7]. Eryptosis is further triggered by a wide variety of xenobiotics and is enhanced in a variety of clinical disorders (i.e., iron deficiency, metabolic syndrome, diabetes, fever and dehydration, sepsis, hemolytic uremic syndrome, mycoplasma infection, myelodysplastic syndrome, phosphate depletion, malaria, sickle-cell anemia, beta-thalassemia, glucose-6-phosphate dehydrogenase-(G6PD)-deficiency, hereditary spherocytosis, paroxysmal nocturnal hemoglobinuria, and Wilson’s disease) [2,5,8,9,10,11,12].

Similar to what already proven for apoptosis of monocytes and renal tubular cells [34,35,36], eryptosis is mostly influenced by blood composition. Voelkl J. et al. reported increased eryptosis in healthy RBCs treated with the plasma of ESRD patient [16], postulating that suicidal erythrocyte death is triggered by complex calcium-phosphate. Furthermore, oxidative stress, inflammation, uremic toxins and cytokines are well-known additional triggers of eryptosis, all factors characterizing the uremic milieu of CKD patients.

In our work, we clearly observed the cytotoxic effect of cytokines (IL-6 and IL-1β) and uremic toxins (urea and p-cresol) on healthy RBCs. This cytotoxic effect was observed both overall and separately for each time point. Every step of eryptosis (cell shrinkage, cell membrane blebbing and scrambling with the exposure of PS on external surface) could be described for treated RBCs. Significantly higher cell death rate was highlighted for RBCs incubated with a higher concentration of molecules. This is consistent with what was previously reported, which is that the percentage of phosphatidylserine exposing erythrocytes is twice as high in patients on dialysis than in healthy individuals [37].

Several reports showed the upregulation of cytokines release [8,38,39] during the inflammatory systemic state and drug-induced immune system dysregulation [40,41] as a cause of the higher level of systemic eryptosis. According to our current results, this hypothesis can be confirmed based on the positive correlation between inflammatory markers and in vitro eryptosis. Similarly, Bester et al. investigated the effect of IL-1β, IL-6 and IL-8 on the structure of erythrocytes and platelets [42], showing that cytokines are responsible for increased hypercoagulability. Importantly, IL-8 strongly affects erythrocytes with visible structural changes to the membrane and eryptosis initiation [42]. Likewise, as C-reactive protein triggers apoptosis in nucleated cells [43], Abed et al. demonstrated its role as inductor of phosphatidylserine translocation in erythrocytes and significantly increased eryptosis [44].

In addition to inflammatory mediators and cytokines, several uremic toxins could affect the RBCs life span. In particular, eryptosis has been proven to be stimulated by molecules derived from self-metabolism and protein catabolism, such as vanadate [45], methylglyoxal [7] and acrolein [46], increased in the case of kidney injury [47]. The accentuated eryptosis in hemodialysis (HD) patients compared to healthy individuals [48,49] has been attributed, in part, to the action of protein-bound uremic toxins (PBUT), such as indoxyl sulfate (IS) [49,50]. Likewise, our in vitro data on urea and p-cresol, both increased in CKD and uremic plasma, confirmed the toxic effect of these molecules on RBCs [50].

To the best of our knowledge, the magnitude of eryptosis and its association with cytokines, such as IL-6 and IL-1β, and uremic toxins, such as urea and p-cresol, has not been yet investigated. In particular, we observed the in vitro ability of IL-6 and IL-1β, and urea and p-cresol to promote eryptosis. In this context, our results support the hypothesis that uremic toxins and inflammatory mediators may play an important role in reduction of RBCs’ life span, mainly in patients affected by chronic kidney disease triggering the specific pathway of eryptosis. From a clinical standpoint, mechanisms and factors influencing eryptosis may provide promising pharmacological target to manage renal anemia, in particular in those patients who develop EPO resistance [51]. Besides the hypoxia-inducible factor prolyl hydroxylase inhibitors, recently approved for clinical use [52], which stimulates hematopoiesis by boosting the synthesis of endogenous EPO, for ESRD patients, the eryptosis pathway could be an interesting goal for future drug development.

To the best of our knowledge, to date there are no available studies about the effect of these molecules on RBCs of CKD patients; therefore, the main novelty of our work lays in the fact that we tested them, analyzing the effect on this specific programmed form of cellular death. Moreover, among those, we chose urea because it is routinely measured in CKD patients and its clinical meaning.

The main limitation of our study is the small sample size. Furthermore, we know that the evaluation of single molecules or single uremic toxins is not enough to describe the complete contribution of the uremic milieu on RBCs. Therefore, our results can be considered to be hypothesis generating, suggesting further investigation for other studies expected to uncover additional triggers and the complete analysis of uremic plasma on the RBCs lifespan.

In conclusion, our data indicate that cytokines and uremic toxins have a harmful effect on RBCs viability and induce eryptosis. Further studies are necessary to validate these results and to associate in vivo abnormal eryptosis and the levels of cytokine and uremic toxins levels in CKD and ESRD patients.

## Figures and Tables

**Figure 1 jcm-11-05329-f001:**
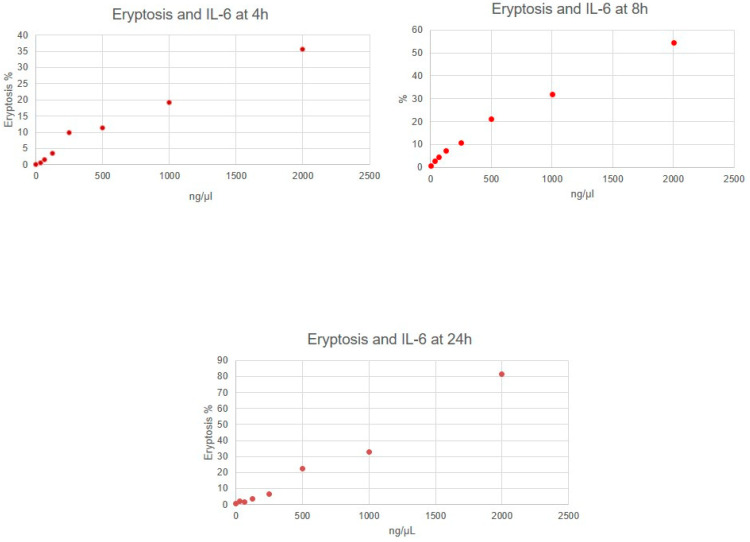
Graphs of eryptosis inducted by IL-6.

**Figure 2 jcm-11-05329-f002:**
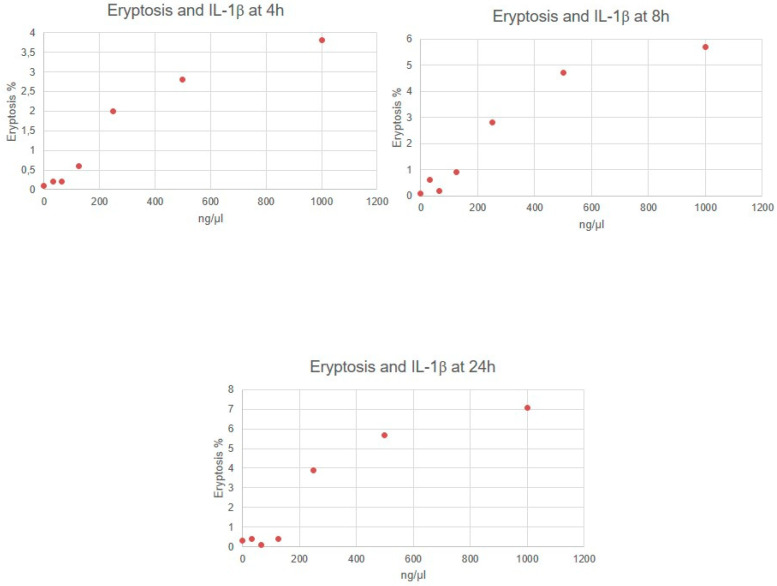
Graphs of eryptosis inducted by IL-1β.

**Figure 3 jcm-11-05329-f003:**
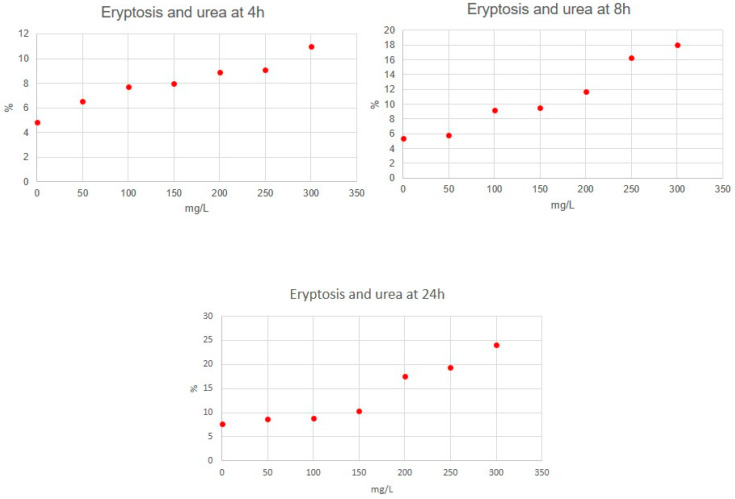
Graphs of eryptosis inducted by urea.

**Figure 4 jcm-11-05329-f004:**
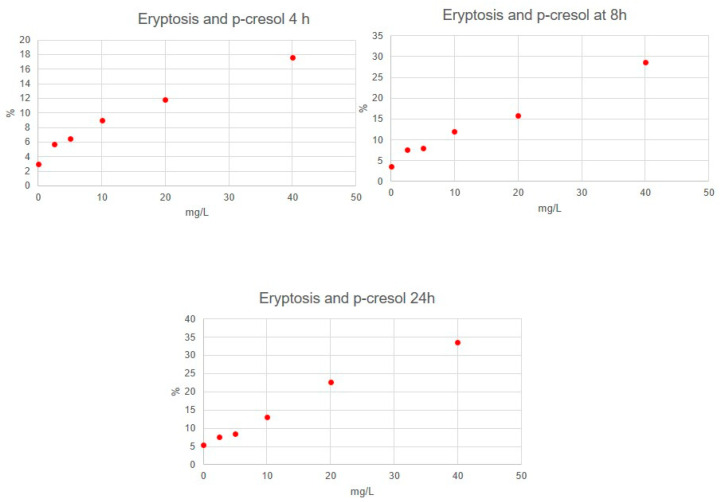
Graphs of eryptosis inducted by p-cresol.

**Table 1 jcm-11-05329-t001:** Correlations between IL-6, IL-1β, urea and p-cresol concentrations and eryptosis level.

	4 h	8 h	24 h
**Spearman’s Correlation**	**p-cresol concentrations**		
	0.95, *p* < 0.01	0.97, *p* < 0.01	0.95, *p* < 0.01
	**urea concentrations**		
	0.92, *p* < 0.01	0.93, *p* < 0.01	0.78, *p* < 0.01
	**IL-6 concentrations**		
	0.98, *p* < 0.01	0.98, *p* < 0.01	0.99, *p* < 0.01
	**IL-1β concentrations**		
	0.45, *p* = 0.05	0.51, *p* = 0.04	0.47, *p* = 0.02

**Table 2 jcm-11-05329-t002:** Comparison between eryptosis percentage at the same concentration of IL-6, IL-1β, urea and p-cresol at 4 h and 24 h.

	Comparison between 4 h and 24 h							
**p-cresol, mg/L**	**40**	**20**	**10**	**5**	**2,5**	**0**		
	17.6 vs. 33.5; *p* < 0.01	11.8 vs. 22.6; *p* < 0.01	9 vs. 13.1; *p* = 0.05	6.5 vs. 8.6; *p* < 0.01	5.7 vs. 7.7; *p* = 0.01	3 vs. 5.4; *p* < 0.01		
**urea, mg/L**	**300**	**250**	**200**	**150**	**100**	**50**	**0**	
	11 vs. 24; *p* = 0.02	9.1 vs. 19.3; *p* = 0.17	8.9 vs. 17.5; *p* = 0.04	8 vs. 10.4; *p* = 0.05	7.7 vs. 8.9; *p* = 0.05	6.5 vs. 8.6; *p* = 0.05	4.8 vs. 7.7; *p* < 0.01	
**IL-6, ng/ µL**	**2000**	**1000**	**500**	**250**	**125**	**65**	**32,5**	**0**
	35.6 vs. 81.3; *p* = 0.01	19.1 vs. 32.6; *p* < 0.01	11.4 vs. 22.6; *p* = 0.01	6.7 vs. 9.8; *p* = 0.05	3.6 vs. 2.6; *p* < 0.01	0.5 vs. 1.5; *p* < 0.01	0.7 vs. 1; *p* = 0.05	0.2 vs. 0.4; *p* = 0.04
**IL-1β, ng/ µL**	**1000**	**500**	**250**	**125**	**65**	**32,5**	**0**	
	3.8 vs. 7; *p* = 0.02	2.6 vs. 5.2; *p* < 0.01	1.8 vs. 3.7; *p* < 0.01	0.8 vs. 0.5; *p* = 0.05	0.3 vs. 0.1; *p* = 0.05	0.2 vs. 0.4; *p* < 0.01	0.1 vs. 0.3; *p* = 0.05	

## Data Availability

All data generated or analyzed during this study are included in this article. Further inquiries can be directed to the corresponding author.

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
