# Peer review of "In Vitro Induction of Eryptosis by Uremic Toxins and Inflammation Mediators in Healthy Red Blood Cells"

_jcm, 2022, doi:10.3390/jcm11185329_

Round 1

Reviewer 1 Report

The aim of this manuscript was to evaluate eryptosis level of healthy RBCs treated with different concentration of urea, p-cresol, IL-6 and IL-1β, comparable to plasmatic level of CKD and uremic patients. Since long ago, healthy RBC incubated in uremic condition led to shorten its survival was confirmed by many literature.

(1)  Just two components of uremic substances are not enough to evaluate uremic condition. Similarly, two components of cytokines are not enough to evaluate specific condition of CKD.

(2)  Only 3 healthy individuals were analyzed in vitro. Author already describes many condition affects RBC survival and eryptosis (i.e. iron deficiency, metabolic syndrome, diabetes, fever, dehydration, sepsis, hemolytic uremic syn and more). How did authors make a choice these volunteers? What characteristic in those volunteer is?

(3)  Cytokines and uremic toxin induced harmful effect on RBCs viability are not novel story unless eryptosis does reveal as novel approach.

(4)  Spearman’s index can not show strong correlation between IL-1β and eryptosis even if p less than 0.05.

Author Response

Manuscript ID: jcm-1861265

Title: IN VITRO INDUCTION OF ERYPTOSIS BY UREMIC TOXINS AND INFLAMMATION
MEDIATORS IN HEALTHY RED BLOOD CELLS

Authors: Grazia Maria Virzì *, Maria Mattiotti, Anna Clementi, Sabrina Milan
Manani, Giovanni Giorgio Battaglia, Claudio Ronco, Monica Zanella

Reviewer 1

Comments and Suggestions for Authors

The aim of this manuscript was to evaluate eryptosis level of healthy RBCs treated with different concentration of urea, p-cresol, IL-6 and IL-1β, comparable to plasmatic level of CKD and uremic patients. Since long ago, healthy RBC incubated in uremic condition led to shorten its survival was confirmed by many literature.

We thank the reviewer for this feedback about our research. The main aim of our work was to investigate specific molecules, in order to clarify their single contribution.

  • Just two components of uremic substances are not enough to evaluate uremic condition. Similarly, two components of cytokines are not enough to evaluate specific condition of CKD.

We thank the reviewer for this observation. We have now specified that our research is a preliminary study. We tested urea, p-cresol, IL-6 and IL-1β in separated experiments and at different concentrations. We know that the use of these uremic components are not enough to evaluate completely uremic condition and uremic plasma composition, but our aim was to evaluate the contribution of some of those in order to better understand the pathogenic contribution of uremic milieu. Thus, our results can be considered as hypothesis-generating, suggesting further investigation. Other studies are expected to uncover additional triggers and complete analysis of uremic plasma.

Only 3 healthy individuals were analyzed in vitro. Author already describes many condition affects RBC survival and eryptosis (i.e. iron deficiency, metabolic syndrome, diabetes, fever, dehydration, sepsis, hemolytic uremic syn and more). How did authors make a choice these volunteers? What characteristic in those volunteer is?

We thank the reviewer for the opportunity to clarify this point. Blood samples were obtained from 3 healthy individuals of age ranging from 30 to 40 years. For any of them any other condition that enhances eryptosis has been excluded. Healthy subjects were recruited in the Blood Center of St. Bortolo Hospital and blood from these subjects were kindly provided by the blood bank of St. Bortolo Hospital. Helathy. Healthy subjects were given a questionarie, a routine medical examination and a biochemical analysis before blood donation.

  • Cytokines and uremic toxin induced harmful effect on RBCs viability are not novel story unless eryptosis does reveal as novel approach.

We thank the reviewer for this observation. At the best of our knowledge, the degree of eryptosis and its association with these specific molecules has not been yet investigated. Thus, the main novelty of our work lays on the fact that we tested effect of these specific molecules molecules on RBCs viability analysing the pathway of this specific programmed form of cellular death.

  • Spearman’s index can not show strong correlation between IL-1βand eryptosis even if p less than 0.05.

We completely agree with the reviewer with this observation, that is why we have now specify with two different sentences for IL-6, urea and p-cresol (very strong correlations with eriptosis) and IL-1β (positive correlations).

We thank the reviewer2 for his/her insightful comments. We believe the manuscript has been improved with the suggested changes. We hope that now our revised manuscript is acceptable for publication. If there is any further information required, please do not hesitate to contact us.

Sincerely,

Reviewer 2 Report

I have reviewed the manuscript jcm-1861265. This study evaluated the eryptosis rate at different concentrations of IL-6, IL-1β, urea, and p-cresol in healthy RBCs. The main issue of the study is that the experimental methodology was not sufficient to simulate the microenvironment of CKD patients and therefore could not correlate the cytokine levels with eryptosis in CKD patients.

This should be addressed further:

1. This study collected whole blood from healthy volunteers and should get an institutional review board statement. Please reveal the approval certificate.

2. There are many kinds of uremic toxins why use p-cresol for this study, but not Indoxyl sulfate, p-cresol sulfate, or others.

3. Eryptosis was determined by flow cytometry. Each sample was collected for a minimum of 100 events, the sample size was small. I suggest increasing to 50,000-100,000 for each sample, which is more applicable to the analysis.

4. This study tested twice for each concentration. I think three separate experiments were better.

5. Please check the sample size enough for analyzing the correlation.

6. The reference style was incorrect (e.g line 36, 39 (2)(3)(4)—> (2-4)). MDPI reference style shows 10 full author names, more than that shown et al. (e.g. Ref. 24 )(https://www.mdpi.com/authors/references)

7. Line 74-76, I suggest using a comma to separate different concentrations.

8. Line 186-189 use different font size.

Author Response

Manuscript ID: jcm-1861265

Title: IN VITRO INDUCTION OF ERYPTOSIS BY UREMIC TOXINS AND INFLAMMATION
MEDIATORS IN HEALTHY RED BLOOD CELLS

Authors: Grazia Maria Virzì *, Maria Mattiotti, Anna Clementi, Sabrina Milan
Manani, Giovanni Giorgio Battaglia, Claudio Ronco, Monica Zanella

Reviewer 2

Comments and Suggestions for Authors

I have reviewed the manuscript jcm-1861265. This study evaluated the eryptosis rate at different concentrations of IL-6, IL-1β, urea, and p-cresol in healthy RBCs. The main issue of the study is that the experimental methodology was not sufficient to simulate the microenvironment of CKD patients and therefore could not correlate the cytokine levels with eryptosis in CKD patients.

We thank the reviewer for the opportunity to explain and clarify this point. This is an in vitro study and results are only preliminary. This is one of the main limitations of the work and we added them into the paper. Our results can be considered as hypothesis-generating, suggesting further investigation. The choice to test urea, p-cresol and cytokines, in separated experiments, has the aim to analyze the contribution of some of the highest represented molecules in plasma of CKD patients on eryptosis, but it is far to create the real complex uremic milieu. Thus, other in vitro studies are expected to discover additional triggers and extend analysis of uremic plasma.

This should be addressed further:

  1. This study collected whole blood from healthy volunteers and should get an institutional review board statement. Please reveal the approval certificate.

We thank the reviewer for his/ her observations. Healthy subjects were recruited in the Blood Center of St. Bortolo Hospital and blood from these subjects were kindly provided by the blood bank of St. Bortolo Hospital. The procedures were in accordance with the Helsinki Declaration. The protocol and consent form were approved by the Ethics Committee of St. Bortolo Hospital (n. 58/17). All patients were informed about the experimental protocol and the objectives of the study before providing informed consent and biological samples.

  1. There are many kinds of uremic toxins why use p-cresol for this study, but not Indoxyl sulfate, p-cresol sulfate, or others.

We thank the reviewer for the opportunity to clarify this point. We choose these molecules because to the best of our knowledge there is not demonstration about their effect on RBCs and eryptosis and its association with cytokines, like IL-6 and IL-1β, and urea and p-cresol has not been yet investigated. Moreover, urea is routinely measured in CKD patients because high level has clinical counterpart. On the contrary, the induction of eryptosis by indosyl sulphate or acrolein were already previously tested from other .

  1. Eryptosis was determined by flow cytometry. Each sample was collected for a minimum of 100 events, the sample size was small. I suggest increasing to 50,000-100,000 for each sample, which is more applicable to the analysis.

We thank the reviewer for the opportunity to clarify this point. There was an error. In fact, we collected a minimum of 100,000 events for each sample

  1. This study tested twice for each concentration. I think three separate experiments were better.

We thank the reviewer for this observation. We now specified that we performed two separated experiments. Each concentration was tested twice both time.

  1. Please check the sample size enough for analyzing the correlation.

We inserted a short point about limits in this preliminary study. We know that the main limitation of this study is the small sample size. Thus, our results can be considered as hypothesis-generating, suggesting further investigation. This could be the starting point for other studies expected to uncover additional triggers and complete analysis of uremic plasma.

  1. The reference style was incorrect (e.g line 36, 39 (2)(3)(4)—> (2-4)). MDPI reference style shows 10 full author names, more than that shown et al. (e.g. Ref. 24 )(https://www.mdpi.com/authors/references)

We corrected Reference List

  1. Line 74-76, I suggest using a comma to separate different concentrations.

We corrected this point.

  1. Line 186-189 use different font size.

We corrected this point.

We thank the reviewer2 for his/her insightful comments. We believe the manuscript has been improved with the suggested changes. We hope that now our revised manuscript is acceptable for publication. If there is any further information required, please do not hesitate to contact us.

Sincerely,

Round 2

Reviewer 1 Report

This munuscript comes along with reviewer's comment precisely.

Author Response

Reviewer 1

This is an awkward description of the data by showing only the spearman correlation and p value. I would like to see the actual values in the figure format in addition to the r and p

We thank the reviewer for the opportunity to clarify this point. We have now inserted 4 different figures (one for each substance) with eryptosis at 4-8-24h (relationship between concentration and eryptosis levels). We removed old figure 1 (redundant data). We left Table 2 with Spearman correlation and p-value. In this way, the figures are simpler  and legible.

What is the actual value of % eryptosis at two different time points for each concentrations? Just showing the p-value does not make any sense

We thank the reviewer for his/ her point. We have now specified percentage values in the table. Please see the new table 2

Comparison between 4h and 24h

p-cresol, mg/L

40

20

10

5

2,5

0

17.6 vs 33.5; p<0.01

11.8 vs 22.6 p<0.01

9 vs 13.1; 0.05

6.5 vs 8.6; p<0.01

5.7 vs 7.7; p=0.01

3 vs 5.4; p<0.01

urea, mg/L

300

250

200

150

100

50

0

11 vs 24, p=0.02

9.1 vs 19.3; p=0.17

8.9 vs 17.5; p=0.04

8 vs 10.4; p=0.05

7.7 vs 8.9; p=0.05

6.5 vs 8.6; p=0.05

4.8 vs 7.7; p=p<0.01

IL-6, ng/µl

2000

1000

500

250

125

65

32,5

0

35.6 vs 81.3; p=0.01

19.1 vs 32.6; p<0.01

 11.4 vs 22.6; p=0.01

6.7 vs 9.8; p=0.05

3.6 vs 2.6; p<0.01

0.5 vs 1.5; p<0.01

0.7 vs 1; p=0.05

0.2 vs 0.4; p=0,04

IL-1β, ng/µl

1000

500

250

125

65

32,5

0

3.8 vs 7; 0.02

2.6 vs 5.2; p<0.01

1.8 vs 3.7; p<0.01

0.8 vs 0.5; p=0.05

0.3 vs 0.1; 0.05

0.2 vs 0.4; p<0.01

0.1 vs 0.3; p=0.05

Reviewer 2 Report

1. The reference list and citation style are still incorrect, please check again. (https://mdpi-res.com/data/mdpi_references_guide_v5.pdf) 

2. Line 128 IL-1B has a strikethrough, please check.

3. Lines 218-220, the text is in bold form, please check.

Author Response

Reviewer 2

  1. The reference list and citation style are still incorrect, please check again. (https://mdpi-res.com/data/mdpi_references_guide_v5.pdf) 

We corrected reference list and citation style.

  1. Line 128 IL-1B has a strikethrough, please check.

We removed IL-1β from line 128 and reported it only in the next sentence.

  1. Lines 218-220, the text is in bold form, please check.

We have now corrected this error
